# TPGDiff: Hierarchical Triple-Prior Guided Diffusion for Image Restoration

Yanjie Tu [1]   Qingsen Yan [* 1 2]   Axi Niu [1]   Jiacong Tang [1]

## Abstract

All-in-one image restoration aims to address diverse degradation types within a unified model. Existing methods typically rely on degradation priors to guide restoration, yet often struggle to reconstruct content in severely degraded regions. Although recent works leverage semantic information to facilitate content generation, integrating it into the shallow layers of diffusion models often disrupts spatial structures (*e.g.*, blurring artifacts). To address this issue, we propose a Triple-Prior Guided Diffusion (TPGDiff) network for unified image restoration. TPGDiff incorporates degradation priors throughout the diffusion trajectory, while introducing structural priors into shallow layers and semantic priors into deep layers, enabling hierarchical and complementary prior guidance for image reconstruction. Specifically, we leverage multi-source structural cues as structural priors to capture fine-grained details and guide shallow layers representations. To complement this design, we further develop a distillation-driven semantic extractor that yields robust semantic priors, ensuring reliable high-level guidance at deep layers even under severe degradations. Furthermore, a degradation extractor is employed to learn degradation-aware priors, enabling stage-adaptive control of the diffusion process across all timesteps. Extensive experiments on both single- and multi-degradation benchmarks demonstrate that TPGDiff achieves superior performance and generalization across diverse restoration scenarios. Our project page is: TPGDiff.

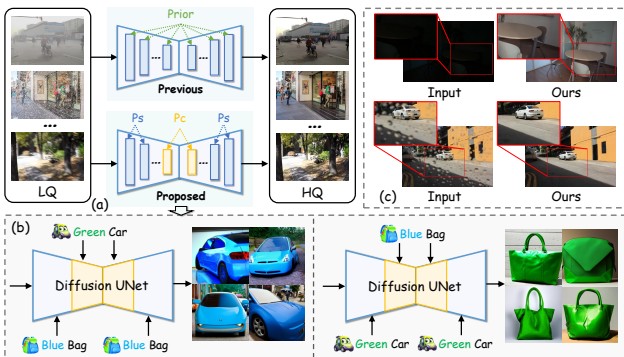

*Figure 1.* (a) Existing methods inject prior information uniformly into the diffusion model, whereas our approach adopts a hierarchical strategy, distributing distinct priors across specific layers of the network. (b) The generation results of diffusion models are largely governed by representations encoded in the deep layers of the network, which play a dominant role in determining the final reconstruction. (c) Visual comparison between the LQ inputs and our restoration results.

## 1. Introduction

Image restoration is a fundamental task in low-level computer vision, aiming to reconstruct high-quality images from degraded low-quality observations. It has been widely applied to various scenarios, including image denoising (Zhang et al., 2018a; Wang et al., 2023b; Zhang et al., 2023b), deblurring (Valanarasu et al., 2022; Tsai et al., 2022; Zhao et al., 2023), desnowing (Liu et al., 2018; Chen et al., 2020; 2021b), dehazing (Song et al., 2023; Wang et al., 2024; Fu et al., 2025), deraining (Chen et al., 2023b;a; Li et al., 2024b), and low-light enhancement (Wu et al., 2023; Xu et al., 2023; Shang et al., 2024). Traditional methods typically focus on addressing a single type of degradation (Chen et al., 2022; Guo et al., 2022; Jin et al., 2023; Li et al., 2023; Lin et al., 2023; Zheng et al., 2023), necessitating the training of separate models for different restoration tasks. However, real-world images often suffer from multiple coupled degradations simultaneously, making task-specific restoration pipelines inefficient and limiting scalability and generalization. To address this issue, recent studies have shifted attention to all-in-one image restoration (Li et al., 2024a; Zhang et al., 2025; Ye et al., 2024; Ai et al., 2024; Luo et al., 2025; Wang et al., 2025a; Zhang et al., 2026; Wang et al., 2025c), which aims to handle

---

[*]Corresponding author [1]School of Computer Science, Northwestern Polytechnical University, Xi'an, China [2]Shenzhen Research Institute of Northwestern Polytechnical University, Shenzhen, China. Correspondence to: Qingsen Yan <qingsenyan@nwpu.edu.cn>.

*Proceedings of the 43rd International Conference on Machine Learning*, Seoul, South Korea. PMLR 306, 2026. Copyright 2026 by the author(s).

multiple degradation types using a single model, thereby improving practicality and general applicability.

Under the all-in-one image restoration framework, different degradation types often exhibit markedly distinct statistical properties and corruption patterns, making it challenging to effectively distinguish diverse degradation scenarios by relying solely on a single restoration model. To alleviate this limitation, a substantial body of existing work (Li et al., 2022; Potlapalli et al., 2023; Wang et al., 2023a; Zhang et al., 2023a; Yao et al., 2024; Hu et al., 2025a) introduces degradation-related information as conditional signals, learning representations associated with degradation types or degradation severity from low-quality images to guide and modulate the restoration behavior under different degradation conditions.

However, degradation information primarily characterizes low-level corruption patterns, and its representation is typically weakly correlated with high-level semantic content. Under complex degradation conditions, relying solely on degradation cues often fails to ensure semantic plausibility of the restored results, leading to issues such as content shift or semantic distortion (Qi et al., 2024). To address this, several recent works (Ma et al., 2023; Luo et al., 2023a; Qu et al., 2024; Wu et al., 2024; Ai et al., 2024; Wei et al., 2025; Kong et al., 2025; Zhang et al., 2025; Wang et al., 2025a; Zhang et al., 2024) leverage pretrained models to extract high-level semantic priors, which are integrated into the restoration network to enforce semantic consistency.

Despite ensuring content consistency, abstract semantic priors often lack the granularity to preserve local geometries. Consequently, their premature integration can compromise spatial integrity, leading to structural blurring or misalignments. Under complex degradation conditions or severe detail loss, relying solely on semantic information often fails to guarantee pixel-level spatial consistency, leading to structural artifacts such as edge discontinuities or geometric distortions. To bridge this gap, several studies incorporate structural priors, such as edge maps or segmentation masks, to assist diffusion models in specific tasks (Zhang et al., 2023c; Mou et al., 2024; Mei et al., 2025). These efforts demonstrate that structural cues provide geometric constraints that are highly complementary to semantic priors. Meanwhile, as the number of incorporated priors increases, effectively exploiting heterogeneous priors remains a key challenge. Most existing methods employ a uniform prior-injection strategy, implicitly assuming that priors elicit similar responses across network layers. However, different layers in diffusion models exhibit distinct modeling behaviors (Voynov et al., 2023; Li et al., 2025c): shallow layers are primarily responsible for encoding local structures and low-level statistics, whereas deeper layers prioritize the synthesis of global semantics and high-level

content, as illustrated in Figure 1.

Motivated by these observations, we leverage the layer-wise characteristics of the UNet backbone in diffusion models to enable feature-prior coordination. Specifically, semantic priors are applied to deep layers to constrain global content, while structural priors are injected into shallow layers to preserve local geometric details. Building on this design, we propose TPGDiff (Triple-Prior Guided Diffusion), a unified image restoration framework that integrates semantic, structural, and degradation priors. By adaptively deploying heterogeneous priors across both the network and the denoising process, TPGDiff achieves a favorable balance between perceptual realism and structural fidelity.

Our contributions can be summarized as follows:

- We propose TPGDiff, a novel prior-guided all-in-one image restoration framework that integrates *semantic*, *structural*, and *degradation* priors to jointly enhance restoration fidelity, perceptual realism, and visual consistency.

- We design a *layer-aware prior coordination* strategy that assigns semantic priors to deep layers and structural priors to shallow layers, enabling effective prior utilization while mitigating semantic drift and preserving local geometric integrity.

- Extensive experiments on both single- and multi-degradation image restoration benchmarks demonstrate that TPGDiff consistently outperforms state-of-the-art methods.

## 2. Related Work

### 2.1. All-in-One Image Restoration

All-in-one image restoration aims to handle multiple image degradations with a single model, thereby avoiding the efficiency and generalization issues caused by training separate models for different tasks. However, due to the substantial differences in the causes and manifestations of various degradations, this problem is inherently more challenging than task-specific image restoration (Cao et al., 2024; Kong et al., 2024; Guo et al., 2024; Li et al., 2025b). To address this challenge, existing methods typically introduce degradation-related information as conditional signals to guide the model in adaptively adjusting its restoration behavior under different degradation scenarios. For instance, Air-Net (Li et al., 2022) employs contrastive learning to obtain discriminative degradation representations, PromptIR (Potlapalli et al., 2023) encodes degradation information using learnable prompts, and NDR (Zhang et al., 2025) achieves dynamic degradation modeling through degradation queries and attention mechanisms. Furthermore, several studies

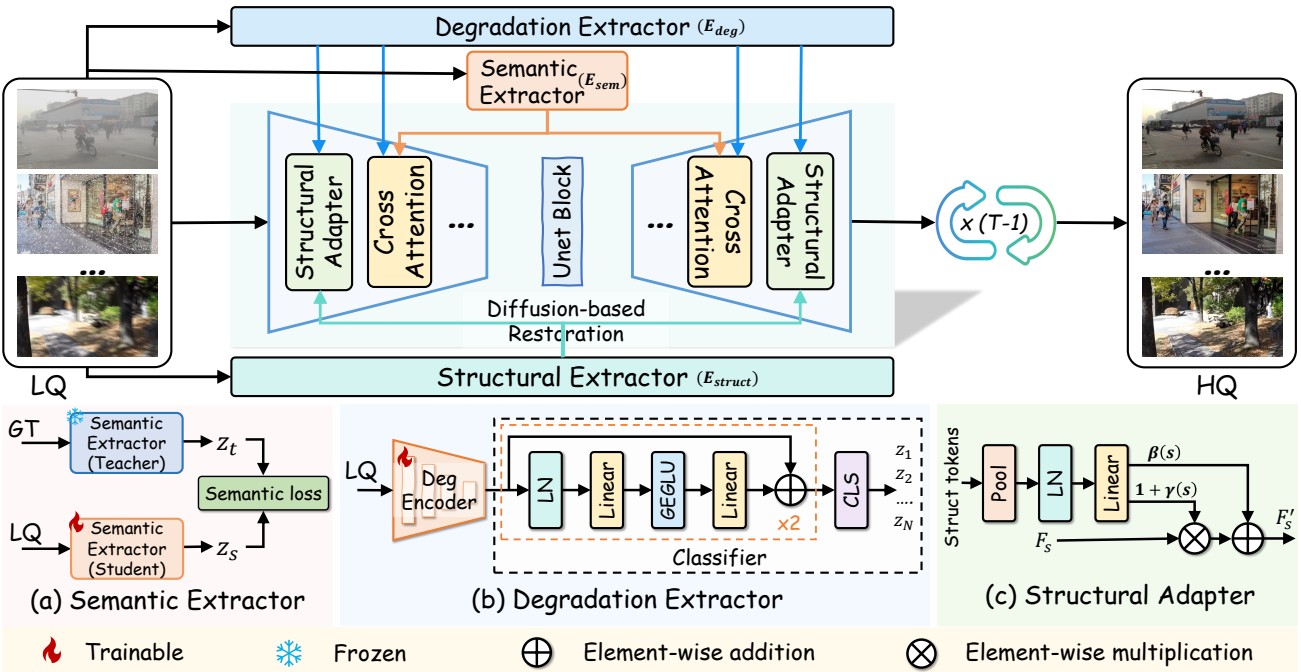

*Figure 2.* Overall architecture of TPGDiff. The framework explicitly models three types of priors from a low-quality input image and integrates them into a diffusion-based restoration network: (a) a *semantic extractor* that learns semantic representations via teacher-student distillation, (b) a *degradation extractor* that captures degradation-related characteristics, and (c) a *structural adapter* that injects structural priors into the diffusion model through an adapter module.

incorporate semantic or content priors to alleviate semantic distortion under severe degradation conditions. For example, Perceive-IR (Zhang et al., 2025) constructs a generic restoration framework via quality-driven prompt learning and difficulty-adaptive perceptual loss, while MaskDCPT (Hu et al., 2025b) enhances generalization through masked image modeling and weakly supervised degradation classification pretraining. Despite these advances, existing methods primarily rely on deterministic mappings, which struggle to model the high-frequency diversity and stochastic uncertainty inherent in complex restoration tasks, often leading to over-smoothed results.

### 2.2. Diffusion-based Image Restoration

In recent years, diffusion models (DMs) have achieved remarkable progress in image generation and editing due to their stable training dynamics and superior generation quality (Ho et al., 2020; Song et al., 2020; Ramesh et al., 2022; Saharia et al., 2022), and have been gradually introduced into image restoration tasks (Ding et al., 2024; Wang et al., 2025b; Zhang et al., 2026). To further improve restoration performance, existing methods typically incorporate conditional information to guide the reverse diffusion process, where the conditions may take the form of degradation-related features, semantic embeddings, or task-specific guidance signals. For example, MPerceiver (Ai et al., 2024) achieves unified diffusion-based image restoration for mul-

tiple degradation tasks through dual-branch conditioning. Diff-Plugin (Liu et al., 2024b) supports multiple image restoration tasks by driving task selection with language instructions and activating different plug-in components. DA-CLIP (Luo et al., 2023a) guides diffusion models to perform multi-task restoration using degradation-type descriptions and semantic prompts as conditions. DiffRes (Wang et al., 2025a) extracts degradation-aware representations from pretrained vision-language models and modulates the diffusion process via feature differencing and adapter-based conditioning. While diffusion models offer stronger generative priors, current approaches typically adopt a unified injection strategy. This overlooks the functional divergence of hierarchical features, failing to resolve the conflicts between global semantics and local structural integrity.

## 3. Method

In this section, we present TPGDiff, a unified diffusion-based framework for all-in-one image restoration. TPGDiff employs a *triple-prior learning* mechanism to capture complementary semantic, structural, and degradation cues. Instead of uniform prior injection, these priors are *layer-aware coordinated* across the UNet backbone, aligning prior types with corresponding feature representations. This design enforces global content consistency while preserving local geometric details under diverse degradation conditions.

### 3.1. Overview of the Proposed Framework

As illustrated in Figure 2, given a low-quality input $x_{\text{LQ}}$, TPGDiff first extracts semantic, structural, and degradation priors using dedicated modules. These priors are layer-aware coordinated within the diffusion model to guide high-fidelity image restoration.

Specifically, a semantic encoder learns semantic representations from $x_{\text{LQ}}$ via a teacher-student distillation scheme,

$$\mathbf{z}_{\text{sem}} = E_{\text{sem}}(x_{\text{LQ}}), \tag{1}$$

while structural and degradation extractors capture corresponding priors,

$$\mathbf{z}_{\text{struct}} = E_{\text{struct}}(x_{\text{LQ}}), \quad \mathbf{z}_{\text{deg}} = E_{\text{deg}}(x_{\text{LQ}}), \tag{2}$$

where details of each prior extractor are presented in subsequent subsections.

For image restoration, we adopt the IR-SDE diffusion (Luo et al., 2023b) framework and recover the high-quality image by solving the reverse stochastic differential equation. Let $\mathbf{x}(t)$ denote the diffusion trajectory at time $t$, the reverse process is given by:

$$\begin{aligned} d\mathbf{x} = &\left[ \theta_t(\boldsymbol{\mu} - \mathbf{x}) - \sigma_t^2 \, s_{\boldsymbol{\theta}}\big(\mathbf{x}, t; \boldsymbol{\mu}, \mathbf{z}_{\text{sem}}, \mathbf{z}_{\text{struct}}, \mathbf{z}_{\text{deg}}\big) \right] dt \\ &+ \sigma_t \, d\hat{\mathbf{w}}. \end{aligned} \tag{3}$$

where $\theta_t$ and $\sigma_t$ are time-dependent coefficients and $\hat{\mathbf{w}}$ denotes the reverse-time Wiener process. The score function $s_{\boldsymbol{\theta}}(\cdot)$ is conditioned on multiple priors, allowing the reverse diffusion process to jointly exploit semantic, structural, and degradation information during progressive denoising.

### 3.2. Semantic Prior Modeling

**Semantic Prior Learning.** Semantic priors are employed to constrain the high-level content consistency of restoration results and should be decoupled from specific degradation information wherever possible. However, when semantic representations are extracted directly from low-quality images, they are often compromised by degradation factors such as noise and blurring (Zhang et al., 2024). To obtain degradation-insensitive semantic representations, we employ a distillation-based semantic learning strategy, transferring stable semantic information from high-quality images through a teacher-student framework. As illustrated in Figure 2(a), we employ a dual-encoder (Radford et al., 2021) distillation framework: the teacher encoder $E_T$ extracts reference semantic features from $x_{\text{GT}}$, while the student encoder $E_{sem}$ aims to recover equivalent representations from the degraded input $x_{\text{LQ}}$

$$\mathbf{z}_t = E_T(x_{\text{GT}}), \quad \mathbf{z}_s = E_{sem}(x_{\text{LQ}}), \tag{4}$$

where $E_T(\cdot)$ is a pretrained encoder with frozen parameters, and $E_{sem}(\cdot)$ is updated during training. We align their semantic spaces via a cosine similarity distillation loss:

$$\mathcal{L}_{\text{sem}} = \frac{1}{B} \sum_{i=1}^{B} \left( 1 - \frac{\mathbf{z}_s^{(i)} \cdot \mathbf{z}_t^{(i)}}{\|\mathbf{z}_s^{(i)}\|_2 \, \|\mathbf{z}_t^{(i)}\|_2} \right), \tag{5}$$

where $\mathbf{z}_s^{(i)}$ and $\mathbf{z}_t^{(i)}$ denote the semantic feature representations of the $i$-th sample produced by the student and teacher encoders, respectively. This constraint encourages the student encoder to distill high-level, degradation-invariant content representations, even under severe low-quality conditions. Once optimized, the features produced by $E_{sem}$ serve as robust semantic priors, anchoring the subsequent restoration process with consistent content guidance.

**Semantic-guided Deep Attention.** Semantic priors primarily constrain the global content consistency of the restored results, making them well suited for the deep modules of the diffusion network, which are intrinsically associated with high-level representations. Guided by this principle, we integrate the semantic prior $\mathbf{z}_s$ as conditional context $\bar{\mathbf{C}}$, enabling it to interact with intermediate feature maps via cross-attention mechanisms within the deep layers of the UNet. Specifically, the intermediate spatial features are denoted as $\bar{\mathbf{X}} \in \mathbb{R}^{L \times D}$, while the projected semantic prior forms a context sequence $\bar{\mathbf{C}} \in \mathbb{R}^{M \times D}$. The output of the cross-attention is defined as:

$$\mathbf{O} = \text{CA}(\bar{\mathbf{X}}, \bar{\mathbf{C}}) = \text{softmax}\left( \frac{\mathbf{Q}\mathbf{K}^\top}{\sqrt{d}} \right) \mathbf{V}. \tag{6}$$

where $\mathbf{Q} = \bar{\mathbf{X}}W_q$, $\mathbf{K} = \bar{\mathbf{C}}W_k$, and $\mathbf{V} = \bar{\mathbf{C}}W_v$, with $W_q$, $W_k$, and $W_v$ denoting linear projection matrices. This design enables each spatial location to adaptively retrieve relevant cues from the global semantic context. Consequently, it effectively mitigates semantic drift during the diffusion sampling process, ensuring object integrity and overall content consistency.

### 3.3. Structural Prior Modeling

**Structural Prior Learning.** Under complex degradation conditions, the geometric structure and boundary morphology of images are often severely compromised, making it difficult to recover fine-grained structural information solely from semantic priors. To this end, we introduce structural priors to capture both local and global geometric cues, thereby facilitating accurate image restoration. As illustrated in Figure 3, we extract three categories of complementary structural cues from the degraded input: depth maps $x^{\text{Dep}}$ to represent global geometric layouts, segmentation maps $x^{\text{Seg}}$ to enforce regional consistency and object boundaries, and Difference-of-Gaussians (DoG) features $x^{\text{DoG}}$ to capture fine-grained contours and high-frequency

details. Depth and segmentation maps are extracted using off-the-shelf pretrained models (Yang et al., 2024; Xie et al., 2021) and remain frozen during training, requiring no additional supervision. To preserve modality discriminability, we adopt a shared lightweight encoder combined with learnable modality embeddings for unified representation learning. Given the $m$-th structural modality input $x^m$, the feature extraction process is formulated as:

$$\mathbf{F}^m = E\big(\phi(x^m) + \mathbf{e}_m\big), m \in \{\mathrm{Dep}, \mathrm{Seg}, \mathrm{DoG}\}, \quad (7)$$

where $\phi(\cdot)$ denotes a lightweight input adaptation module, and $\mathbf{e}_m$ is the corresponding modality embedding. Subsequently, the feature maps are flattened into token sequences:

$$\mathbf{T}^{(m)} = \mathrm{Flat}(\mathbf{F}^m), \mathbf{T}_{\mathrm{M}} = [\mathbf{T}^{(d)}, \mathbf{T}^{(s)}, \mathbf{T}^{(g)}], \quad (8)$$

Since multi-cue structural tokens typically exhibit high redundancy and information overlap, we aim to compress them into a more compact representation. Inspired by (Wang et al., 2020; Jaegle et al., 2021; Mei et al., 2025), we introduce a Structural Token Aggregator (STA) as shown in Figure 3. Specifically, a set of learnable latent tokens $\mathbf{L} \in \mathbb{R}^{N \times D}$ is employed to actively aggregate essential information from the full set of structural tokens via cross-attention:

$$\tilde{\mathbf{L}} = \mathrm{CA}(\mathbf{Q} = \mathbf{L}, \mathbf{K} = \mathbf{T}_{\mathrm{M}}, \mathbf{V} = \mathbf{T}_{\mathrm{M}}), \quad (9)$$

Subsequently, linear projection followed by self-attention is applied to enhance token-wise consistency and further reduce inter-modal redundancy, yielding the final structural prior:

$$\mathbf{z}_{\mathrm{struct}} = \mathrm{SA}\big(\mathrm{Linear}(\tilde{\mathbf{L}})\big), \quad (10)$$

By jointly leveraging depth, semantic segmentation, and DoG cues, this approach suppresses erroneous edges and spurious artifacts, yielding consistent and coherent geometric representations even under severe degradation. The resulting structural prior provides reliable constraints to guide subsequent restoration.

**Structural Adapter.** The structural prior primarily characterizes the spatial organization and geometric relationships within an image, which is particularly critical for preserving fine-grained structures. To this end, we introduce a Structural Adapter, as shown in Figure 2(c), which injects the structural prior into the shallow layer feature channels of the UNet in a lightweight and parameter-efficient manner, thereby directly constraining local features that are sensitive to structural information. Specifically, the structural tokens are first aggregated into a global structural representation $\mathbf{s}$ via an aggregation operator $P(\cdot)$. Subsequently, a parameterized mapping function $F(\cdot)$ is employed to predict channel-wise modulation parameters:

$$[\boldsymbol{\gamma}(\mathbf{s}), \boldsymbol{\beta}(\mathbf{s})] = F(P(\mathbf{s})), \quad (11)$$

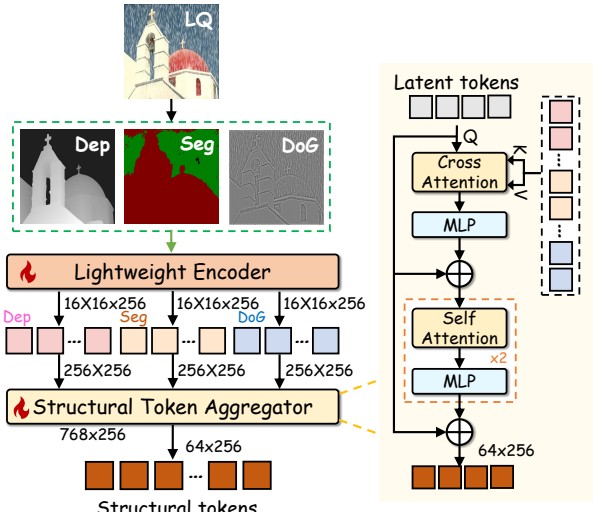

*Figure 3.* Design of the proposed structural extractor with multi-cue token aggregation. It aggregates heterogeneous tokens to learn unified structural representations for downstream image restoration tasks.

Given the shallow layer features $\mathbf{F}_s$, the structural modulation is formulated as:

$$\mathbf{F}'_s = \mathbf{F}_s \odot \big(1 + \boldsymbol{\gamma}(\mathbf{s})\big) + \boldsymbol{\beta}(\mathbf{s}). \quad (12)$$

By modulating features with structural guidance, the Structural Adapter enforces spatial constraints that prevent the loss of fine-grained structural details, mitigating common artifacts like over-smoothing during the denoising process.

### 3.4. Degradation Prior Modeling

**Degradation Prior Learning.** While semantic and structural priors ensure the fidelity and consistency of the restored results, effective restoration under complex degradations still requires accurate degradation information. However, in all-in-one image restoration, since degradation and content cues reside in distinct representational subspaces, their coupling often leads to degradation-content ambiguity, destabilizing degradation awareness and resulting in inaccurate adaptive restoration or residual artifacts in complex scenarios (Jiang et al., 2025). To address this, we design an independent Degradation Extractor, which learns content-agnostic degradation representations in a dedicated representation subspace, thereby enabling more robust and accurate degradation-aware restoration.

As illustrated in Figure 2(b), given a low-quality input $x_{\mathrm{LQ}} \in \mathbb{R}^{H \times W \times 3}$, we employ a pretrained CLIP (Radford et al., 2021) visual encoder $E_v$ to extract global features. Leveraging its extensive pretraining on diverse datasets, CLIP provides robust representations that encapsulate rich

global semantic and quality-related information.

$$\mathbf{F}_g = E_{deg}(x_{\mathrm{LQ}}) \in \mathbb{R}^D, \qquad (13)$$

Based on these features, we introduce a degradation classifier for discriminative supervised learning. The classifier outputs a degradation logit vector $\mathbf{z}_i \in \mathbb{R}^N$, where $N$ denotes the number of predefined degradation categories. During training, we adopt the cross-entropy loss with label smoothing:

$$\mathcal{L}_{\mathrm{deg}} = -\frac{1}{B} \sum_{i=1}^{B} \sum_{c=1}^{N} q_{i,c} \log \frac{\exp(z_{i,c})}{\sum_{j=1}^{N} \exp(z_{i,j})}, \qquad (14)$$

where $q_{i,c} = (1 - \varepsilon)\mathbb{I}[c = y_i] + \varepsilon/N$ with $\varepsilon = 0.01$. Here, $z_{i,c}$ denotes the logit of the $i$-th sample for class $c$, and $y_i$ is the corresponding degradation label.

Notably, the degradation classifier is only used during training to guide the model to learn degradation-related discriminative representations. At inference time, the degradation classifier is discarded, and only the continuous features learned by the encoder are retained as the degradation prior $\mathbf{z}_{\mathrm{deg}}$, which is subsequently used for conditional modulation in the diffusion-based restoration process.

**Degradation-aware Time Modulation.** Degradation information is typically global in nature and is closely correlated with the noise evolution across the diffusion trajectory. Combining degradation priors with temporal conditions helps the model adaptively adjust its denoising strategy under different noise scales (Luo et al., 2023a). Therefore, we encode the degradation prior together with the time embedding, allowing it to participate in conditional modulation throughout the entire diffusion trajectory. Specifically, given a diffusion timestep $\tau$, its time embedding is defined as:

$$\mathbf{t} = \psi(\tau) \in \mathbb{R}^{D_t}, \qquad (15)$$

where $\psi(\cdot)$ denotes the time embedding mapping function. We project the degradation prior $\mathbf{z}_{\mathrm{deg}}$ into the time embedding space and modulate it via a learnable temporal adapter:

$$\mathbf{t}' = \mathbf{t} + \phi\big(\mathrm{softmax}(W_d \mathbf{z}_{\mathrm{deg}}) \odot \mathbf{P}\big). \qquad (16)$$

where $W_d$ denotes a learnable projection matrix, $\mathbf{P} \in \mathbb{R}^{D_t}$ is a learnable prompt vector, and $\phi(\cdot)$ represents a prompt transformation function. This mechanism enables degradation information to be incorporated into temporal conditioning, thereby guiding the model to dynamically adjust its denoising behavior with respect to degradation characteristics across different noise stages.

## 4. Experiments

### 4.1. Experimental Setups

**Datasets.** We evaluate our method on multiple degradation-specific, multi-task, and unknown-task image restoration

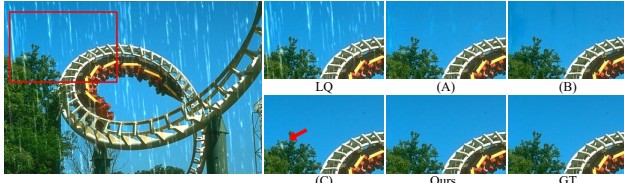

*Figure 4.* Visual ablation study on prior components. Each prior plays a distinct and indispensable role, where their synergy ensures both geometric integrity and semantic consistency.

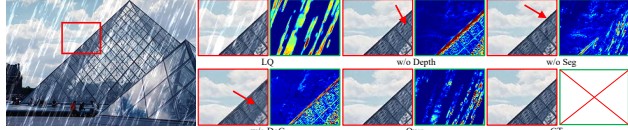

*Figure 5.* Visual ablation study on structural priors. The restored patches and pixel-wise error maps show that removing different structural priors leads to distinct types of structural errors.

benchmarks. For each degradation type, we collected data from corresponding datasets: Low light: LOL-v1 (Wei et al., 2018) and LOL-v2 (Yang et al., 2021); Non-Homogeneous: NH-HAZE (Ancuti et al., 2020); Moire: RDNet (Yue et al., 2022); Rain: Rain100L and Rain200L (Yang et al., 2017); Snow: Snow100K-L (Liu et al., 2018) and WeatherBench (Guan et al., 2025); Haze: RESIDE6K (Qin et al., 2020) and SOTS (Li et al., 2018); Raindrops: Raindrop (Qian et al., 2018); Defocus blur: DPDD (Abuolaim & Brown, 2020); Clouds: Recloud (Ning et al., 2025); Noise: BSD400 (Martin et al., 2001), WED (Ma et al., 2016), and CBSD68 (Martin et al., 2001); Motion blur: Go-Pro (Nah et al., 2017); Unknown-degradation: POLED and TOLED (Zhou et al., 2021).

**Implementation Details.** The proposed method is trained in a two-stage manner. In the first stage, we use a batch size of 512 with an initial learning rate of $2 \times 10^{-5}$. This stage is trained for 30 epochs on a single NVIDIA A100 GPU. In the second stage, the batch size is set to 16, and the input images are randomly cropped to $256 \times 256$ for data augmentation. The learning rate is fixed at $2 \times 10^{-5}$, the diffusion noise level is set to $\sigma = 50$, and the number of denoising steps is set to 100. In this stage, we adopt the AdamW (Loshchilov & Hutter, 2017) optimizer and employ a cosine learning rate decay strategy. Training is conducted on two NVIDIA A100 GPUs for a total of 700K iterations.

**Evaluation Metrics.** To comprehensively evaluate the quantitative performance of different methods, we employ multiple commonly used and widely recognized evaluation metrics. At the pixel level, PSNR and SSIM (Wang et al., 2004) are employed to measure the fidelity between reconstructed results and reference images. For perceptual quality, FID (Heusel et al., 2017) and LPIPS (Zhang et al., 2018b) are respectively utilized to characterize differences in fea-

*Table 1.* Quantitative comparison between our method and other state-of-the-art approaches on nine different degradation-specific datasets. ↑ represents the bigger the better, and ↓ represents the smaller the better. Best and second-best results are highlighted in red and blue, respectively.

| Method | Low-Light Enhancement (LOL-v2) | | | | Method | NH Dehazing (NH-HAZE) | | | | Method | Demoire (RDNet) | | | |
|---|---|---|---|---|---|---|---|---|---|---|---|---|---|---|
| | PSNR↑ | SSIM↑ | FID↓ | LPIPS↓ | | PSNR↑ | SSIM↑ | MUSIQ↑ | LPIPS↓ | | PSNR↑ | SSIM↑ | FID↓ | LPIPS↓ |
| Retinexformer | 22.80 | 0.840 | 62.45 | 0.169 | Transweather | 11.58 | 0.411 | 40.22 | 0.692 | SwinIR | 24.89 | 0.888 | 28.73 | 0.100 |
| URetinex-Net | 19.84 | 0.824 | 52.38 | 0.237 | AutoDIR | 12.71 | 0.477 | 46.29 | 0.612 | RDNet | 26.16 | 0.941 | 23.64 | 0.091 |
| Restormer | 20.77 | 0.851 | 57.04 | 0.115 | DiffUIR | 11.39 | 0.422 | 47.77 | 0.651 | InstructIR | 24.69 | 0.843 | 32.18 | 0.111 |
| NAFNet | 18.04 | 0.827 | 54.25 | 0.147 | InstructIR | 12.24 | 0.498 | 54.80 | 0.530 | UniRestore | 24.06 | 0.819 | 45.28 | 0.155 |
| AirNet | 19.69 | 0.821 | 55.43 | 0.151 | AgenticIR | 12.20 | 0.450 | 37.47 | 0.675 | FoundIR | 24.71 | 0.876 | 32.49 | 0.107 |
| PromptIR | 21.23 | 0.860 | 53.92 | 0.145 | X-Restormer | 11.36 | 0.413 | 44.28 | 0.665 | DCPT | 24.18 | 0.815 | 31.38 | 0.159 |
| MPerceiver | 22.16 | 0.848 | 45.90 | 0.130 | FoundIR-v2 | 17.00 | 0.462 | 65.62 | 0.334 | MaskDCPT | 25.21 | 0.942 | 24.41 | 0.095 |
| DA-CLIP | 21.76 | 0.762 | 48.23 | 0.134 | DA-CLIP | 12.35 | 0.466 | 49.73 | 0.590 | DA-CLIP | 24.75 | 0.826 | 38.71 | 0.134 |
| **Ours** | 24.77 | 0.880 | 46.02 | 0.106 | **Ours** | 18.14 | 0.645 | 58.43 | 0.268 | **Ours** | 26.40 | 0.910 | 22.07 | 0.089 |

| Method | Deraining (Rain100L) | | | | Method | Desnowing (WeatherBench) | | | | Method | Dehazing (RESIDE) | | | |
|---|---|---|---|---|---|---|---|---|---|---|---|---|---|---|
| | PSNR↑ | SSIM↑ | FID↓ | LPIPS↓ | | PSNR↑ | SSIM↑ | MUSIQ↑ | LPIPS↓ | | PSNR↑ | SSIM↑ | FID↓ | LPIPS↓ |
| JORDER | 36.61 | 0.974 | 14.66 | 0.028 | PromptIR | 21.54 | 0.777 | 45.20 | 0.242 | GCANet | 26.59 | 0.935 | 11.52 | 0.052 |
| PRENET | 37.48 | 0.979 | 10.98 | 0.020 | TransWeather | 20.94 | 0.758 | 44.25 | 0.253 | GridDehazeNet | 25.86 | 0.944 | 10.62 | 0.048 |
| MAXIM | 38.06 | 0.977 | 19.06 | 0.048 | DiffUIR | 21.68 | 0.780 | 45.22 | 0.240 | DehazeFormer | 30.29 | 0.964 | 7.58 | 0.045 |
| IR-SDE | 38.30 | 0.981 | 7.94 | 0.014 | X-Restormer | 21.57 | 0.781 | 46.04 | 0.250 | MAXIM | 29.12 | 0.932 | 8.12 | 0.043 |
| GOUB | 39.79 | 0.983 | 5.18 | 0.010 | AgenticIR | 20.35 | 0.730 | 50.87 | 0.283 | IR-SDE | 25.25 | 0.906 | 8.33 | 0.060 |
| Perceiver-IR | 38.41 | 0.984 | - | - | FoundIR | 21.57 | 0.779 | 45.72 | 0.241 | AdaIR | 30.87 | 0.975 | - | - |
| VLU-Net | 38.60 | 0.984 | - | - | FoundIR-v2 | 23.15 | 0.706 | 60.15 | 0.254 | DiffRes | 30.87 | 0.970 | 5.08 | 0.023 |
| DA-CLIP | 37.02 | 0.978 | 8.96 | 0.012 | DA-CLIP | 21.59 | 0.773 | 45.39 | 0.236 | DA-CLIP | 30.16 | 0.936 | 5.52 | 0.030 |
| **Ours** | 40.12 | 0.986 | 3.47 | 0.007 | **Ours** | 28.17 | 0.836 | 50.63 | 0.104 | **Ours** | 32.87 | 0.979 | 4.69 | 0.021 |

| Method | Raindrop Removal (RainDrop) | | | | Method | Defocus Deblur (DPDD) | | | | Method | Declouding (RS-Cloud) | | | |
|---|---|---|---|---|---|---|---|---|---|---|---|---|---|---|
| | PSNR↑ | SSIM↑ | FID↓ | LPIPS↓ | | PSNR↑ | SSIM↑ | FID↓ | LPIPS↓ | | PSNR↑ | SSIM↑ | MUSIQ↑ | LPIPS↓ |
| AttGAN | 31.59 | 0.917 | 22.38 | 0.058 | NRKNet | 26.11 | 0.817 | 43.96 | 0.223 | PromptIR | 11.35 | 0.772 | 41.83 | 0.271 |
| Quanetal | 31.37 | 0.918 | 30.56 | 0.065 | DRBNet | 25.72 | 0.806 | 39.37 | 0.182 | DiffUIR | 13.82 | 0.814 | 38.44 | 0.260 |
| WeatherDiff | 27.06 | 0.847 | 49.06 | 0.089 | InstructIR | 23.84 | 0.746 | 84.88 | 0.329 | InstructIR | 14.46 | 0.865 | 47.00 | 0.163 |
| RDDM | 27.07 | 0.805 | 51.86 | 0.102 | UniRestore | 22.91 | 0.724 | 91.59 | 0.364 | X-Restormer | 11.48 | 0.751 | 35.99 | 0.325 |
| DiffUIR | 26.88 | 0.817 | 55.84 | 0.114 | FoundIR | 23.45 | 0.742 | 89.21 | 0.358 | AgenticIR | 17.80 | 0.799 | 34.90 | 0.257 |
| DiffRes | 33.70 | 0.939 | 19.65 | 0.047 | DCPT | 25.68 | 0.816 | 42.59 | 0.216 | FoundIR | 11.71 | 0.753 | 35.40 | 0.322 |
| IRBridge | 26.91 | 0.813 | 49.15 | 0.098 | MaskDCPT | 25.64 | 0.809 | 38.49 | 0.183 | FoundIR-v2 | 22.06 | 0.828 | 44.37 | 0.125 |
| DA-CLIP | 30.75 | 0.892 | 24.31 | 0.061 | DA-CLIP | 23.55 | 0.747 | 67.54 | 0.288 | DA-CLIP | 16.43 | 0.803 | 35.73 | 0.238 |
| **Ours** | 32.29 | 0.923 | 19.16 | 0.046 | **Ours** | 26.85 | 0.816 | 42.04 | 0.144 | **Ours** | 38.37 | 0.954 | 42.97 | 0.037 |

*Table 2.* Comparison under unknown tasks setting on the POLED (Zhou et al., 2021) dataset. ↑ represents the bigger the better, and ↓ represents the smaller the better. Best and second-best results are highlighted in red and blue, respectively.

| Method | POLED | | |
|---|---|---|---|
| | PSNR ↑ | SSIM ↑ | LPIPS ↓ |
| HINet (Chen et al., 2021a) | 11.52 | 0.436 | 0.831 |
| MPRNet (Zamir et al., 2021) | 8.34 | 0.365 | 0.798 |
| SwinIR (Liang et al., 2021) | 6.89 | 0.301 | 0.852 |
| DGUNet (Mou et al., 2022) | 8.88 | 0.391 | 0.810 |
| NAFNet (Chen et al., 2022) | 10.83 | 0.416 | 0.794 |
| MIRV2 (Zamir et al., 2022b) | 10.27 | 0.425 | 0.722 |
| Restormer (Zamir et al., 2022a) | 9.04 | 0.399 | 0.742 |
| RDDM (Liu et al., 2024a) | 15.58 | 0.398 | 0.544 |
| DL (Fan et al., 2019) | 13.92 | 0.449 | 0.756 |
| TAPE (Liu et al., 2022) | 7.90 | 0.219 | 0.799 |
| AirNet (Li et al., 2022) | 7.53 | 0.350 | 0.820 |
| DA-CLIP (Luo et al., 2023a) | 14.91 | 0.475 | 0.739 |
| DiffUIR (Zheng et al., 2024) | 15.62 | 0.424 | 0.505 |
| DeepSN-Net (Deng et al., 2025) | 10.20 | 0.411 | 0.534 |
| MAIR (Li et al., 2025a) | 11.07 | 0.423 | 0.529 |
| AWRaCLe (Rajagopalan & Patel, 2025) | 11.21 | 0.431 | 0.513 |
| **Ours** | 15.64 | 0.511 | 0.644 |

*Table 3.* Comparison under unknown tasks setting on the TOLED (Zhou et al., 2021) dataset. ↑ represents the bigger the better, and ↓ represents the smaller the better. Best and second-best results are highlighted in red and blue, respectively.

| Method | TOLED | | |
|---|---|---|---|
| | PSNR ↑ | SSIM ↑ | LPIPS ↓ |
| HINet (Chen et al., 2021a) | 13.84 | 0.559 | 0.448 |
| MPRNet (Zamir et al., 2021) | 24.69 | 0.707 | 0.347 |
| SwinIR (Liang et al., 2021) | 17.72 | 0.661 | 0.419 |
| DGUNet (Mou et al., 2022) | 19.67 | 0.627 | 0.384 |
| NAFNet (Chen et al., 2022) | 26.89 | 0.774 | 0.346 |
| MIRV2 (Zamir et al., 2022b) | 21.86 | 0.620 | 0.408 |
| Restormer (Zamir et al., 2022a) | 20.98 | 0.632 | 0.360 |
| RDDM (Liu et al., 2024a) | 23.48 | 0.639 | 0.383 |
| DL (Fan et al., 2019) | 21.23 | 0.656 | 0.434 |
| TAPE (Liu et al., 2022) | 17.61 | 0.583 | 0.520 |
| AirNet (Li et al., 2022) | 14.58 | 0.609 | 0.445 |
| DA-CLIP (Luo et al., 2023a) | 15.74 | 0.606 | 0.472 |
| DiffUIR (Zheng et al., 2024) | 29.55 | 0.887 | 0.281 |
| DeepSN-Net (Deng et al., 2025) | 17.39 | 0.576 | 0.303 |
| MAIR (Li et al., 2025a) | 23.64 | 0.770 | 0.271 |
| AWRaCLe (Rajagopalan & Patel, 2025) | 10.54 | 0.495 | 0.331 |
| **Ours** | 32.70 | 0.899 | 0.145 |

ture distribution distance and perceived similarity between generated results and real images. Additionally, MUSIQ (Ke et al., 2021) is introduced to evaluate overall image quality.

### 4.2. Comparison with State-of-the-art Methods

**Single-task Image Restoration Results.** We evaluate the proposed method across nine representative single-

degradation restoration benchmarks. Table 1 summarizes the quantitative comparisons between TPGDiff, task-specific baselines, and state-of-the-art all-in-one restoration approaches. These benchmarks cover diverse degradation scenarios, providing a comprehensive evaluation of the model's restoration capability. The results demonstrate that TPGDiff consistently achieves superior or competitive performance across the majority of tasks and met-

*Table 4.* Quantitative comparison between our method and other state-of-the-art approaches on five restoration tasks. ↑ represents the bigger the better. Best and second-best results are highlighted in red and blue, respectively.

| METHOD | LOW-LIGHT | | DEBLURRING | | DERAINING | | DEHAZING | | DENOISING | | AVERAGE | |
|---|---|---|---|---|---|---|---|---|---|---|---|---|
| | PSNR↑ | SSIM↑ | PSNR↑ | SSIM↑ | PSNR↑ | SSIM↑ | PSNR↑ | SSIM↑ | PSNR↑ | SSIM↑ | PSNR↑ | SSIM↑ |
| AIRNET (LI ET AL., 2022) | 18.18 | 0.735 | 24.35 | 0.781 | 32.98 | 0.951 | 21.04 | 0.884 | 30.91 | 0.882 | 25.49 | 0.846 |
| IR-SDE (LUO ET AL., 2023B) | 20.07 | 0.780 | 26.34 | 0.800 | 34.12 | 0.951 | 24.56 | 0.940 | 30.89 | 0.865 | 27.20 | 0.867 |
| PROMPTIR (POTLAPALLI ET AL., 2023) | 22.89 | 0.829 | 27.93 | 0.851 | 36.17 | 0.970 | 30.41 | 0.972 | 31.20 | 0.885 | 29.72 | 0.901 |
| DA-CLIP (LUO ET AL., 2023A) | 21.66 | 0.828 | 27.31 | 0.838 | 35.65 | 0.962 | 29.78 | 0.968 | 30.93 | 0.885 | 29.07 | 0.896 |
| DIFFUIR (ZHENG ET AL., 2024) | 22.32 | 0.826 | 27.50 | 0.845 | 35.98 | 0.968 | 29.47 | 0.965 | 31.02 | 0.885 | 29.25 | 0.898 |
| INSTRUCTIR (CONDE ET AL., 2024) | 20.70 | 0.820 | 26.65 | 0.810 | 35.58 | 0.967 | 25.20 | 0.938 | 31.09 | 0.883 | 27.84 | 0.884 |
| FOUNDIR (LI ET AL., 2025B) | 12.81 | 0.642 | 25.87 | 0.801 | 29.39 | 0.918 | 19.74 | 0.878 | 24.45 | 0.637 | 22.45 | 0.775 |
| DCPT (HU ET AL., 2025A) | 20.38 | 0.836 | 26.42 | 0.807 | 35.70 | 0.974 | 28.67 | 0.973 | 31.16 | 0.882 | 28.47 | 0.894 |
| POOLNET (CUI ET AL., 2025) | 22.66 | 0.841 | 27.66 | 0.844 | 37.85 | 0.981 | 30.25 | 0.977 | 31.24 | 0.887 | 29.93 | 0.906 |
| TUR (WU ET AL., 2025) | 17.88 | 0.772 | 26.13 | 0.801 | 33.95 | 0.962 | 27.59 | 0.954 | 30.93 | 0.875 | 27.30 | 0.873 |
| VLU-NET (ZENG ET AL., 2025) | 23.00 | 0.852 | 27.46 | 0.840 | 38.54 | 0.982 | 30.84 | 0.980 | 31.43 | 0.891 | 30.11 | 0.905 |
| **OURS** | 24.52 | 0.857 | 28.94 | 0.863 | 38.05 | 0.982 | 30.90 | 0.987 | 30.58 | 0.853 | 30.60 | 0.908 |

*Figure 6.* Visual comparison with other all-in-one image restoration methods on image denoising, low-light enhancement, image deraining, and image deblurring tasks. Zoom in for a better view.

*Table 5.* **Ablation study on prior types.** We evaluate the impact of different prior types on guiding the recovery process.

| MODEL | METHOD | | | METRICS | | |
|---|---|---|---|---|---|---|
| | DEG. | SEM. | STRUC. | PSNR ↑ | SSIM ↑ | LPIPS ↓ |
| (A) | ✗ | ✗ | ✗ | 31.63 | 0.912 | 0.060 |
| (B) | ✓ | ✗ | ✗ | 31.93 | 0.912 | 0.060 |
| (C) | ✓ | ✓ | ✗ | 32.61 | 0.920 | 0.049 |
| OURS | ✓ | ✓ | ✓ | **32.63** | **0.921** | **0.045** |

rics. Notably, TPGDiff performs well not only on common weather-related degradations but also on more challenging degradation-specific and domain-specific scenarios, such as image demoireing, defocus deblurring, and remote-sensing image declouding. This validates the effectiveness of our prior-guided diffusion framework in single-degradation settings and underscores its robust generalization across diverse restoration challenges.

**Multi-task All-in-one Image Restoration Results.** We further evaluate TPGDiff's performance in multi-degradation joint training scenarios. Specifically, we select five typical degradation types for joint training and compare them with multiple integrated image restoration methods. This setting is more challenging than single-task restoration, as the model must distinguish degradation-specific patterns while maintaining a shared restoration capability across heterogeneous tasks. As shown in Table 4, TPGDiff achieves optimal or suboptimal results across all quantitative metrics, demonstrating stronger multitask unified modeling capabilities. This indicates that TPGDiff can effectively balance task-specific degradation removal and task-agnostic image reconstruction within a unified framework. Further visualization results, as depicted in Figure 6, demonstrate that

TPGDiff more stably restores image structure and details across diverse degradation scenarios, exhibiting superior robustness and visual consistency. Additional visualization results and analyses are presented in the appendix.

**Unknown Task Generalization.** To evaluate the generalization ability of the model under unknown complex degradations, we conduct quantitative experiments on the POLED and TOLED (Zhou et al., 2021) datasets. Captured by under-display cameras, these datasets exhibit high-resolution characteristics and real-world degradation artifacts, making them suitable for evaluating cross-domain robustness. As shown in Tables 2 and 3, the proposed method outperforms existing approaches across multiple metrics, demonstrating strong generalization ability under unseen degradation scenarios. This suggests that the proposed framework can effectively transfer its learned restoration priors to degradation distributions that are substantially different from those observed during training.

### 4.3. Ablation Study

In this section, we conduct ablation experiments to evaluate the impact of individual components in TPGDiff on the overall restoration performance. Unless otherwise specified, the ablation experiments are conducted on the Rain200L (Yang et al., 2017) and Snow100K-L (Liu et al., 2018) datasets, and the reported metrics are averaged over both tasks.

**Effectiveness of Different Types of Priors.** To evaluate the contribution of each prior, we conduct a progressive ablation study, as reported in Table 5. Starting from the baseline, the incorporation of degradation and semantic priors leads to improvements in both pixel-level accuracy and perceptual quality. With the further introduction of the structural prior, all evaluation metrics are consistently optimized. The visual comparisons in Figure 4 confirm the necessity and synergy of each prior component.

**Effectiveness of Prior Input Positions.** To further investigate the sensitivity differences of diffusion model layers to various priors, we conduct an alignment study on prior injection positions. As shown in Table 6, the model achieves optimal performance when semantic priors are applied to deep layers and structural priors are injected into shallow layers. These results validate our prior-position alignment strategy, indicating that matching prior types with layer-specific representation properties is crucial for effective restoration.

**Effectiveness of Structural Prior Components.** To further analyze the structural prior, we remove depth, segmentation, and DoG information separately while keeping other settings unchanged. These experiments are conducted on Rain200L (Yang et al., 2017) and LOL-v2 (Yang et al., 2021). As shown in Table 7, removing any component

*Table 6.* **Ablation study on input positions.** We compare the four combinations of injecting semantic and structural priors at deep versus shallow layers.

| SEMANTIC PRIOR | | STRUCTURAL PRIOR | | METRICS | | |
| --- | --- | --- | --- | --- | --- | --- |
| DEEP | SHALLOW | DEEP | SHALLOW | PSNR ↑ | SSIM ↑ | LPIPS ↓ |
| ✗ | ✗ | ✓ | ✓ | 31.73 | 0.912 | 0.061 |
| ✓ | ✓ | ✗ | ✗ | 32.38 | 0.918 | 0.051 |
| ✗ | ✓ | ✓ | ✗ | 32.18 | 0.915 | 0.052 |
| ✓ | ✗ | ✗ | ✓ | **32.63** | **0.921** | **0.045** |
| ✓ | ✓ | ✓ | ✓ | 31.51 | 0.910 | 0.061 |

*Table 7.* **Ablation study on structural prior components.** We evaluate the contribution of Depth, Seg, and DoG priors by removing each component separately.

| MODEL | METHOD | | | METRICS | | |
| --- | --- | --- | --- | --- | --- | --- |
| | DEPTH | SEG | DoG | PSNR ↑ | SSIM ↑ | LPIPS ↓ |
| W/O DEPTH | ✗ | ✓ | ✓ | 29.00 | 0.903 | 0.066 |
| W/O SEG | ✓ | ✗ | ✓ | 29.22 | 0.893 | 0.067 |
| W/O DoG | ✓ | ✓ | ✗ | 28.94 | 0.895 | 0.068 |
| OURS | ✓ | ✓ | ✓ | **29.46** | **0.905** | **0.066** |

degrades performance, indicating that the three structural cues are complementary and jointly provide a more complete structural representation. The corresponding visual comparisons are shown in Figure 5.

## 5. Conclusion

In this paper, we presented TPGDiff, a novel Triple-Prior Guided Diffusion framework designed for unified image restoration. The core of our approach lies in the hierarchical and complementary coordination of multi-faceted priors. By integrating multi-source structural cues into shallow layers and distillation-driven semantic priors into deep layers, TPGDiff successfully bridges the gap between geometric precision and contextual plausibility. Furthermore, the inclusion of degradation-aware priors ensures stage-adaptive guidance throughout the entire diffusion trajectory. Extensive evaluations confirm that our tri-prior integration effectively mitigates structural distortion and semantic drift, setting a new benchmark for robustness and fidelity in all-in-one restoration tasks.

## Impact Statement

This research aims to address image restoration challenges and does not involve any ethical issues. The study process entails neither subjective evaluations nor the use of private data, relying solely on publicly available datasets for experimentation.

## Acknowledgements

This work is supported by the NSFC of China under Grant 62301432 and 62306240.

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
