# OpenReview forum: "TPGDiff : Hierarchical Triple-Prior Guided Diffusion for Image Restoration"
_ICML.cc/2026/Conference — ICML 2026 regular_

### Official Review · Reviewer_7a9d · 2026-03-04

**Soundness:** 3
**Presentation:** 3
**Significance:** 2
**Originality:** 3
**Overall Recommendation:** 5
**Confidence:** 4

**Summary:**

This work investigates the different functions of the priors when injected into different layers and proposes a triple-prior guided diffusion network for better restoration. TPGDiff adopts structural and semantic priors into shallow and deep layers to fully utilize the priors. The semantic extractor is distilled from a teacher model for better generalization. Through the denoising process, TPGDiff also inputs degradation priors across all the steps.

**Compliance With Llm Reviewing Policy:**

Affirmed.

**Final Justification:**

The authors provide sufficient evidence of their contribution and address my concerns. Although the overall framework is complex and possesses limited capability for generalization, the insightful technical contributions are valuable. Furthermore, I notice that the reviewers with negative scores both claim that most of their concerns have been solved. Therefore, I finally decide to raise my score to Accept recommendation.

**Key Questions For Authors:**

see questions in weakness.

**Limitations:**

1. Discuss the specific contribution of each prior.
2. The generalization capability across different tasks.

**Strengths And Weaknesses:**

S1. The authors investigate the generation results of diffusion models with different settings of prior injection. The comparison motivates the design of using multiple kinds of priors.

S2. The discussion of previous works is comprehensive. Through the issues and observation of existing progress, the authors introduce the utilization of multiple priors. The presentation of the method is clear, and the illustrations of the figures and captions are easy to follow.

S3. The experimental results are sufficient to demonstrate the effectiveness of TPGDiff. The ablation studies are well organized and clearly show the contribution of each technique.

W1. In Fig 1, there is no comparison of visual results (only prior design) between previous methods and TPGDiff, which limits the clarity of how the previous design produces poor outputs.

W2. The real contribution of each prior is not clear. The size of Fig 4 is too small. And the semantic prior may not align well with natural language (the example in Fig 1). How do you ensure the priors act as the correct roles?

---

> ### Author Rebuttal · Authors · 2026-03-30
>
> **For W1:** We thank the reviewer for the valuable suggestion. We have revised Fig. 1 (Fig. 2 in the link: https://anonymous-author2026.github.io/tpgd-re/) by adding a visual comparison of restoration results between previous methods and TPGDiff. In addition, we will replace Fig. 1 in the revised manuscript to more clearly present the differences in the final restoration results produced by different methods.
>
> **For W2:** We thank the reviewer for the valuable comments. We have reorganized the layout of Fig. 4 (Fig. 3 in the link: https://anonymous-author2026.github.io/tpgd-re/) and enlarged cropped key regions to present the detailed changes more intuitively. Meanwhile, we supplement two additional ablation settings with different injection positions, (a) and (e), as shown in Table 1, to more clearly analyze the role of different priors at different levels. The related experiments are all trained under the same settings on the Rain200L and Snow100K-L datasets.
>
> In addition, we would like to clarify that the semantic prior in this paper refers to high-level category/region semantic consistency information and does not require strict alignment with natural language. We have conducted a visualization analysis of feature responses at different levels (Fig. 1 in the link: https://anonymous-author2026.github.io/tpgd-re/). Shallow features respond more strongly to edges and textures, whereas the activations of deep features are more concentrated on the main object regions, demonstrating stronger semantic relevance and regional selectivity. Our method constrains the functional positions of different priors through hierarchical injection, and we will further strengthen this part of the analysis in the revised manuscript.
> |Table1|Sem-Deep|Sem-Shallow|Stru-Deep|Stru-Shallow|PSNR↑|SSIM↑|LPIPS↓|
> |---|---|---|---|---|---:|---:|---:|
> |(a)|×|×|√|√|31.73|0.912|0.061|
> |(b)|√|√|×|×|32.38|0.918|0.051|
> |(c)|×|√|√|×|32.18|0.915|0.052|
> |(d)|√|×|×|√|32.63|0.921|0.045|
> |(e)|√|√|√|√|31.51|0.910|0.061|

---

> > ### Author Rebuttal · Reviewer_7a9d · 2026-04-01
> >
> > Thank you for the detailed rebuttal and the additional visual materials provided in the link. My initial concerns regarding the visual comparisons and the specific roles of the priors have been well addressed by the new figures and the ablation study.
> > Although my personal concerns are resolved, I will hold off on updating my score for now. During the discussion phase, I will mainly focus on the concerns raised by the two reviewers who gave negative ratings, and I look forward to their replies. I will make my final decision and adjust my score based on those discussions.

---

> > > ### Author Response · Authors · 2026-04-05
> > >
> > > Thank you for further confirming that our response has addressed your concerns. We also fully understand your intention to make your final judgment in conjunction with the subsequent discussion, and we appreciate your willingness to continue paying attention to the issues raised by the other reviewers. We will participate in the subsequent discussion seriously, and we also hope that the clarifications and evidence provided in the paper and rebuttal can be advantageous to the relevant discussion. Thank you again for your careful and thoughtful review comments.

---

### Official Review · Reviewer_4w4G · 2026-03-05

**Soundness:** 3
**Presentation:** 4
**Significance:** 3
**Originality:** 3
**Overall Recommendation:** 5
**Confidence:** 5

**Summary:**

This paper proposes a TPGDiff to simultaneously improve the structural clarity and semantic fidelity of the results. Specifically, TPGDiff first injects structural features into the shallow layer of the diffusion model and then injects semantic features into the deep layer. In addition, a degenerate feature is introduced into the model. Experimental results show that TPGDiff achieves excellent performance.

**Compliance With Llm Reviewing Policy:**

Affirmed.

**Final Justification:**

The author has addressed essentially all questions, including those from other reviewers. The method proposed by the author offers a new approach to image restoration tasks. Based on this, I believe this paper can be accepted.

**Key Questions For Authors:**

1. In Section 3.3, the authors extracted three complementary structural cues from the degraded input: depth map, segmentation map, and Difference of Gaussian (DoG) features. The depth map and segmentation map were extracted using off-the-shelf pre-trained models. However, how were the DoG features obtained?

2. Since the image is degraded by different degradation, extracted structural information may be inaccurate. Did the authors consider extracting some structures that might not be affected by degradation? For example, the dark channel prior mentioned in DCP and the residual channel prior mentioned in SPDNet.

[1] DCP: Single Image Haze Removal using Dark Channel Prior [2] SPDNet: Structure-Preserving Deraining with Residue Channel Prior Guidance

**Limitations:**

1. Was the diffusion model used in TPGDiff trained from scratch, or did it use a pre-trained diffusion model such as SD2? The authors could clarify this in the paper.
2. Some related work may be missing:
[1] DCP: Single Image Haze Removal using Dark Channel Prior
[2] SPDNet: Structure-Preserving Deraining with Residue Channel Prior Guidance
[3] Textual prompt guided image restoration
[4] Rain o'er me: Synthesizing real rain to derain with data distillation

**Strengths And Weaknesses:**

The motivation of this paper is clear, the proposed solution is well designed, and the paper is easy to read.

---

> ### Author Rebuttal · Authors · 2026-03-30
>
> **For Question 1:** We thank the reviewer for the question. Different from depth maps and segmentation maps, DoG features are directly computed from the degraded input image and do not rely on any additional pretrained model. Specifically, we first convert the input image into grayscale, then use 3×3 and 9×9 average pooling to approximate Gaussian blurring at different scales, and obtain the DoG response through differencing so as to highlight local edges and structural changes. Subsequently, it is normalized to [-1, 1] and fed into the structural encoder as supplementary structural cues. We will supplement these implementation details in the revised manuscript to avoid ambiguity.
>
> In addition, we supplement ablation experiments, as shown in Table 1, where depth, seg, and DoG information are removed separately and the models are trained under the same settings on Rain200L and LOL-v2.
> |Table1|Depth|Seg|DoG|PSNR↑|SSIM↑|LPIPS↓|
> |---|:---:|:---:|:---:|---:|---:|---:|
> |w/o Depth|×|√|√|29.00|0.903|0.066|
> |w/o Seg|√|×|√|29.22|0.893|0.067|
> |w/o DoG|√|√|×|28.94|0.895|0.068|
> |Ours|√|√|√|29.46|0.905|0.066|
>
> **For Question 2:** We thank the reviewer for the valuable suggestion. In complex degradation scenarios, directly extracting structural information from degraded images may introduce bias. To alleviate this issue, we jointly use Depth, Seg, and DoG to construct the structural prior rather than relying on a single cue. As for the RCP in DCP and SPDNet, we consider them inspiring. However, as demonstrated by our experimental results in Table 2, these two types of priors are usually dependent on specific degradation assumptions and therefore are difficult to directly serve as general structural priors under the unified restoration setting of this work. We will supplement this discussion in the revised manuscript and regard learning more degradation-invariant structural priors as a direction for future work.
>
> (Note: since the official code of DCP is not publicly available, we use the GitHub implementation with the most stars.)
> |Table2|PSNR↑|SSIM↑|LPIPS↓|
> |---|---|---|---|
> |RCP|28.90|0.894|0.069|
> |DCP|28.43|0.891|0.075|
> |Ours|29.46|0.905|0.066|

---

> > ### Author Rebuttal · Reviewer_4w4G · 2026-04-03
> >
> > Thank you for the author's reply. The provided ablation experiments demonstrate that structural cues can effectively aid image restoration. Furthermore, these structures are more helpful for unified image restoration than some specific priors such as DCP and RCP. Although my problem has been solved, there are two negative scores. During the discussion phase, I will mainly focus on the concerns raised by the two reviewers who gave negative ratings, and I look forward to their replies. I will make my final decision and adjust my score based on those discussions.

---

> > > ### Author Response · Authors · 2026-04-05
> > >
> > > Thank you for further confirming that our response has addressed your concerns. We also sincerely appreciate your willingness to continue paying attention to the issues raised by the other reviewers during the discussion phase. We will carefully respond to the relevant discussion, and we also hope that the supplementary experiments and explanations provided can offer valuable reference for the subsequent exchange. Thank you again for your careful reading and feedback.

---

### Official Review · Reviewer_WbKF · 2026-03-06

**Soundness:** 3
**Presentation:** 3
**Significance:** 2
**Originality:** 2
**Overall Recommendation:** 3
**Confidence:** 3

**Summary:**

This paper proposes TPGDiff, a diffusion-model framework for image restoration tasks. The authors argue that existing diffusion-based restoration methods tend to suffer from semantic misalignment and loss of structural details in scenarios involving multiple degradations. Therefore, they attempt to constrain the diffusion process by introducing multiple types of prior information. The method designs three types of priors: a semantic prior, a structural prior, and a degradation prior. The semantic prior learns degradation-invariant semantic features through a teacher–student distillation framework, and integrates them into the deep features of the UNet via cross-attention, thereby enhancing high-level semantic consistency.

**Compliance With Llm Reviewing Policy:**

Affirmed.

**Key Questions For Authors:**

(1) Regarding the hierarchical injection strategy, the authors are encouraged to provide further analysis or visualization results, such as the responses of features at different levels to semantic or structural information, in order to help readers better understand the rationale behind the current design choice.
(2) For the structural prior module, it would be helpful to include more fine-grained ablation studies, such as using depth, segmentation, or DoG information separately, so as to more clearly evaluate the individual contribution of each component and the actual benefit of their combination.
(3) Considering that the method relies on multiple additional models, the authors are encouraged to supplement the paper with an analysis of the computational overhead, such as inference time, number of model parameters, or overall computational complexity, so as to provide a more comprehensive assessment of the method’s efficiency in practical applications.
(4) If possible, it would also be beneficial to include more challenging experimental settings, such as unseen degradation types or complex real-world degradation scenarios, in order to further validate the model’s generalization ability.

**Limitations:**

(1) The method relies on multiple external models to extract semantic and structural information, such as depth estimation and semantic segmentation models. In practical applications, the prediction quality of these models may be affected by the quality of the input images or shifts in data distribution, which may in turn indirectly influence the final restoration results.
(2) Since multiple types of prior information need to be integrated, TPGDiff introduces greater overall system complexity and computational overhead compared with a basic diffusion model. In resource-constrained or real-time application scenarios, this additional cost may become a limiting factor.
(3) At present, the experiments in the paper are mainly conducted on datasets with known degradation types. For more complex real-world degradations or unseen degradation types, the generalization ability of the model still requires further verification.

**Strengths And Weaknesses:**

Strengths:
1.The paper proposes a diffusion framework guided by multiple prior information to address the problem of multi-degradation image restoration. By integrating semantic, structural, and degradation priors into the diffusion model, the method attempts to constrain the restoration process from multiple perspectives. Overall, the design idea is relatively clear and intuitive.
2.The model architecture is relatively well developed, and the interface relationships between the various prior modules and the diffusion backbone are also fairly clear. For example, the semantic prior is fused through cross-attention, while the structural prior is injected into the shallow layers of the network via feature modulation, making the overall architecture reasonably implementable.
Weaknesses:
1.The paper proposes injecting different priors at different network levels, but the current explanation for this design still relies mainly on experimental results. Although Table 4 provides an ablation comparison of prior injection at different levels, the paper lacks further analysis explaining why semantic information is better suited to deep features, whereas structural information is more appropriate for shallow features.
2.The structural prior consists of three components: a depth map, a semantic segmentation map, and a DoG edge map. Although the authors state in the method section that these types of information are complementary, the experiments do not provide a fine-grained decomposition of their individual contributions. As a result, it remains unclear what specific role each type of structural information plays in the overall performance improvement.
3.The method relies on multiple additional models to extract prior information, such as a depth estimation model, a semantic segmentation model, and a semantic distillation network. This increases the overall system complexity to some extent, yet the paper does not provide an analysis of inference time, additional parameter size, or computational overhead. In addition, conditioning the diffusion model on multiple priors simultaneously may make the model heavily dependent on auxiliary information. At present, the experiments are mainly conducted under known degradation settings, and the robustness of the model under unseen degradation types or more complex real-world degradation scenarios still requires further validation.

---

> ### Author Rebuttal · Authors · 2026-03-30
>
> **For Weakness-1 and Question-1:** We thank the reviewer for the valuable suggestion. We have supplemented a visualization analysis of features at different levels. The results show (Fig. 1 in the link: https://anonymous-author2026.github.io/tpgd-re/) that shallow features respond more strongly to edges and textures, whereas the activations of deep features are more concentrated on the main object regions, demonstrating stronger semantic relevance and regional selectivity.
>
> To further verify the effectiveness of the hierarchical injection strategy, we supplement two additional ablation settings, (a) and (e). As shown in Table 1, if semantic and structural priors are injected into all layers simultaneously, the performance instead degrades; in contrast, the hierarchical design that introduces structural priors into shallow layers and semantic priors into deep layers achieves the best results. This indicates that the roles of the two types of priors are not symmetric across different levels: unified injection introduces representation interference. The related experiments are all trained under the same settings on the Rain200L and Snow100K-L datasets.
>
> (Note: If scheme (e) injects semantic information into the shallow layers through an attention-based manner, it leads to OOM under the same experimental setting. Therefore, the actual implementation adopts an adapter-based manner consistent with the structural prior.)
> |Table1|Sem-Deep|Sem-Shallow|Stru-Deep|Stru-Shallow|PSNR↑|SSIM↑|LPIPS↓|
> |---|---|---|---|---|---:|---:|---:|
> |(a)|×|×|√|√|31.73|0.912|0.061|
> |(b)|√|√|×|×|32.38|0.918|0.051|
> |(c)|×|√|√|×|32.18|0.915|0.052|
> |(d)|√|×|×|√|32.63|0.921|0.045|
> |(e)|√|√|√|√|31.51|0.910|0.061|
>
> **For Weakness-2 and Question-2:** To further analyze the role of each component in the structural prior, we supplement ablation experiments in which depth, seg, and DoG information are removed separately while keeping all other settings unchanged, and the models are trained under the same settings on the Rain200L and LOL-v2 datasets. As shown in Table 2, removing any structural prior leads to performance degradation. This indicates that all three types of structural information contribute positively to the final performance and that their effects are not redundant. Since they provide structural constraints at different levels, they can form a more complete structural representation, thereby yielding the best restoration performance. We will further clarify the role of each component in the revised manuscript.
> |Table2|Depth|Seg|DoG|PSNR↑|SSIM↑|LPIPS↓|
> |-----------|:-----:|:---:|:---:|------:|------:|-------:|
> |w/o Depth|×|√|√|29.00|0.903|0.066|
> |w/o Seg|√ |×|√|29.22|0.893|0.067|
> |w/o DoG|√ |√|×|28.94|0.895|0.068|
> |Ours|√|√|√|29.46|0.905|0.066|
>
> **For Weakness-3 and Question-3:** We have supplemented a complexity comparison under a unified experimental setting. All results are evaluated on an NVIDIA Tesla A100 (40G). As shown in Table 3, although TPGDiff, as a diffusion-based method, incurs a relatively higher inference cost, its computational overhead remains within a reasonable range among similar diffusion-based methods while yielding superior restoration performance and generalization ability.
> |Table3|Params|FLOPs|Inference Time|
> |---|---|---|---|
> |Restormer|26M|155G|0.25s|
> |AirNet|9M|301G|0.57s|
> |ProRes|371M|139G|0.21s|
> |PromptIR|36M|172G|0.36s|
> |InstructIR|33M|292G|0.33s|
> |CyclicPrompt|181M|230G|0.47s|
> |DA-RCOT|41M|262G|0.63s|
> |MoCE-IR|25M|142G|0.27s|
> |VLUNet|123M|191G|0.53s|
> |IR-SDE|137M|379G|4.61s|
> |DA-CLIP|295M|138G|4.24s|
> |Ours|355M|189G|4.18s|
>
> **For Question-4:** We thank the reviewer for the valuable suggestion. More challenging experiments are extremely important for validating generalization ability. In addition to single-degradation and multi-degradation tasks, this paper has already been evaluated on unknown-degradation datasets, but this part was originally placed in the appendix and was not sufficiently presented in the main text. In the revised manuscript, we will move this part to the main text and provide further analysis. In addition, we further include experiments on a real denoising scenario (SIDD), as shown in Table 4, to verify the effectiveness and generalization ability of the model under real degradation scenarios.
> |Table4|PSNR↑|SSIM↑|LPIPS↓|
> |---|---|---|---|
> |MPRNet|39.71|0.958|0.200|
> |UFORMER|39.89|0.960|0.198|
> |RESTORMER|40.02|0.967|0.195|
> |ART|39.99|0.960|0.189|
> |AIRNET|38.32|0.945|0.134|
> |PROMPTIR|39.52|0.954|0.198|
> |DA-CLIP|34.04|0.824|0.186|
> |MPERCEIVER|39.96|0.959|0.191|
> |VARFORMER|40.13|0.978|-|
> |OURS|40.22|0.964|0.073|

---

> > ### Author Rebuttal · Reviewer_WbKF · 2026-04-03
> >
> > Thank you for the detailed response. Most of my concerns have been well addressed.

---

> > > ### Author Response · Authors · 2026-04-06
> > >
> > > Thank you for the reviewer’s further feedback. We sincerely apologize that the related issues were only partially addressed in the previous version. Following the reviewer’s suggestions, we have supplemented the analysis (https://anonymous-author2026.github.io/tpgdiff-re2/). The related explanations are as follows:
> > >
> > > **1. Rationale of the Hierarchical Prior Injection Strategy.**
> > >
> > > To further clarify the motivation of the hierarchical design, we additionally visualized representative shallow and deep features at the intermediate reverse iteration step t = 20 (Figure 4). The results show that shallow features mainly respond to local high-frequency structures, while deep features exhibit stronger aggregation in the main object regions, reflecting more evident semantic relevance. In contrast, uniform injection tends to blur this distinction. Therefore, injecting structural priors into shallow layers and semantic priors into deep layers is more consistent with the characteristics of hierarchical network representations.
> > >
> > > **2. Robustness of Semantic Priors to Image Quality.**
> > >
> > > We thank the reviewer for pointing out this important issue. Our semantic priors are not simply extracted directly from degraded images but are learned from high-quality semantic anchors through a teacher-student distillation framework (T-HQ vs. S-LQ). As shown in Figure 1, if semantic representations are directly extracted from degraded images (T-HQ vs. T-LQ), there is an obvious shift from the clean semantic space. In contrast, the distilled student representations (T-HQ vs. S-LQ) exhibit higher consistency on the diagonal entries. This indicates that the distilled semantic representations can more stably approximate the clean semantic anchors provided by HQ images, thereby alleviating, to some extent, the influence of degraded input quality on semantic priors.
> > >
> > > **3. Specific Roles of Different Structural Priors.**
> > >
> > > For the specific roles of the three structural priors, we have supplemented a fine-grained visualization analysis (Figure 3). Combined with the pixel-wise error maps, it can be observed that these structural priors help reduce structural errors at different levels, thereby better preserving the geometric consistency and local detail fidelity of the restored results. This phenomenon is also consistent with the ablation study results: removing any one of the structural priors leads to performance degradation, indicating that the three are not simply redundant.
> > >
> > > **4. Generalization Ability under Unknown Degradation Scenarios.**
> > >
> > > We will further present experimental results on the unknown degradation tasks POLED and TOLED [1] in the main text (Table 1) in order to more clearly demonstrate the model’s generalization ability under unknown degradation scenarios and further strengthen this part of the analysis.
> > >
> > > |Table 1|POLED PSNR↑|POLED SSIM↑|POLED LPIPS↓|TOLED PSNR↑|TOLED SSIM↑|TOLED LPIPS↓|
> > > |---|---|---|---|---|---|---|
> > > |HINet|11.52|0.436|0.831|13.84|0.559|0.448|
> > > |MPRNet|8.34|0.365|0.798|24.69|0.707|0.347|
> > > |SwinIR|6.89|0.301|0.852|17.72|0.661|0.419|
> > > |DGUNet|8.88|0.391|0.810|19.67|0.627|0.384|
> > > |NAFNet|10.83|0.416|0.794|26.89|0.774|0.346|
> > > |MIRV2|10.27|0.425|0.722|21.86|0.620|0.408|
> > > |Restormer|9.04|0.399|0.742|20.98|0.632|0.360|
> > > |RDDM|15.58|0.398|0.544|23.48|0.639|0.383|
> > > |DL|13.92|0.449|0.756|21.23|0.656|0.434|
> > > |TAPE|7.90|0.219|0.799|17.61|0.583|0.520|
> > > |AirNet|7.53|0.350|0.820|14.58|0.609|0.445|
> > > |DA-CLIP|14.91|0.475|0.739|15.74|0.606|0.472|
> > > |DiffUIR|15.62|0.424|0.505|29.55|0.887|0.281|
> > > |DeepSN-Net|10.20|0.411|0.534|17.39|0.576|0.303|
> > > |MAIR|11.07|0.423|0.529|23.64|0.770|0.271|
> > > |AWRACLE|11.21|0.431|0.513|10.54|0.495|0.331|
> > > |Ours|15.64|0.511|0.644|32.70|0.899|0.145|
> > >
> > >
> > > Once again, we sincerely thank the reviewer for the constructive comments. These suggestions have been very helpful in further improving the analysis and presentation of our paper.
> > >
> > > [1] Image restoration for under-display camera.

---

### Official Review · Reviewer_5RxG · 2026-03-15

**Soundness:** 3
**Presentation:** 3
**Significance:** 2
**Originality:** 2
**Overall Recommendation:** 3
**Confidence:** 4

**Summary:**

The authors propose **TPGDiff**, an all-in-one image restoration framework based on diffusion models. The core idea is a "**Triple Prior**" guidance mechanism that extracts semantic, structural, and degradation priors from the input image and injects them into different layers of a U-Net backbone. Specifically, structural priors are applied to shallow layers, semantic priors to deep layers, and degradation priors are used to globally modulate the diffusion timesteps. The method is evaluated on several image restoration benchmarks under both single-task and multi-task joint training settings.

**Compliance With Llm Reviewing Policy:**

Affirmed.

**Final Justification:**

After reading the rebuttal, I appreciate the authors’ clarifications and the additional experiments. The response improves the presentation and partially addresses some empirical concerns, particularly by supplementing the prior-injection ablation and providing efficiency statistics. However, it does not sufficiently change my overall assessment. Most importantly, the rebuttal effectively narrows the scope of the paper from restoration under genuinely coupled degradations to all-in-one multi-task restoration. In my view, this leaves the main issue intact: the original manuscript motivates the problem using coupled degradations, while the method and evaluation are built around single-label degradation modeling and predominantly single-dominant degradation benchmarks. This mismatch remains central to my concern. The added ablation strengthens the case for the preferred injection strategy, but the broader contribution still reads primarily as a careful architectural coordination of heterogeneous priors rather than a stronger methodological advance for ICML. I also remain unconvinced on the efficiency/performance trade-off. In the manuscript, TPGDiff evaluates its 5D all-in-one setting on the standard LOL-v1, GoPro, Rain100L, SOTS/RESIDE, and BSD68 suite, and reports an average PSNR of 30.60 dB. By comparison, MoCE-IR（25M） reports essentially the same average performance on the closely related AIO-5 setting (30.58 dB), while using a much lighter architecture. In the rebuttal, the authors additionally report 355M parameters and 189G FLOPs for TPGDiff, which makes the empirical advantage over recent efficient baselines difficult to view as substantial enough to justify the added complexity. Overall, the rebuttal improves clarity, but it does not sufficiently change the core reasons behind my original recommendation, so I am maintaining my Weak Reject.

**Key Questions For Authors:**

1. How does the degradation classifier (Eq. 14), which uses mutually exclusive cross-entropy loss, handle an image that actually contains "multiple coupled degradations" (e.g., Rain + Snow + Low-light) simultaneously?
2. Please provide a comprehensive computational analysis (total parameters, FLOPs, and inference time per image) comparing TPGDiff with key baselines (e.g., VLU-NET, MOCE-IR [1]).
3. In Table 4, what would the quantitative results be if the model injected both semantic and structural priors into both shallow and deep layers simultaneously?

**Limitations:**

The methodology of the paper involves the use of AI image generation technology, which could give rise to certain ethical concerns.

**Strengths And Weaknesses:**

### **Strengths**
1. **Extensive Experimental Coverage:** The authors demonstrate the effectiveness of the proposed method across 9 single-task datasets and a 5-task joint training benchmark. The quantitative results on these specific benchmarks appear competitive.
2. **Clear Exposition:** The overall pipeline is well-illustrated, and the paper is relatively easy to follow. It clearly describes how the three types of priors are extracted and where they are integrated into the network.

### **Weaknesses**

1. **Significant Misalignment Between Motivation and Execution (Coupling vs. All-in-One):**
   While the paper intends to address a wide range of real-world degradations, it fundamentally conflates the concepts of "**multi-coupled degradations**" and "**multi-task all-in-one restoration**."
   * In the introduction (Lines 41-43), the paper explicitly motivates the need for handling "simultaneously occurring multi-coupled degradations" (e.g., an image affected by rain, low-light, and noise concurrently).
   * However, the proposed degradation extractor (Section 3.4, Eq. 14) utilizes a **cross-entropy loss** with single-class labels, which inherently forces the model to assume that degradations are mutually exclusive.
   * Furthermore, the primary multi-task experiments in Table 2 only co-train the model on datasets where each image contains only *one* type of degradation. The model acts as a "**task router**" rather than a true solver for coupled degradations. The method is neither designed for nor primarily evaluated against the problem it claims to solve in the motivation.

2. **Unjustified Architectural Complexity and Missing Efficiency Metrics:**
   The proposed pipeline relies on an ensemble of numerous auxiliary extractors: a semantic student encoder, a frozen depth estimator, a frozen segmentation model, a Difference-of-Gaussian (DoG) module, a structural token aggregator, and a pre-trained CLIP vision encoder—all to condition a 100-step diffusion reverse process.
   * Despite the claim of being "parameter-efficient" (Line 257), the paper entirely lacks any analysis of **parameter count, MACs/FLOPs, or inference latency**.
   * Given that the entire structural prior module yields only a marginal gain of **0.02 dB PSNR** (Table 3, Model C vs. Ours), this heavy pipeline appears highly redundant and computationally impractical.
   * Additionally, the performance is not particularly outstanding when compared to [1] (MOCE-IR), which is not cited.

3. **Flawed and Incomplete Core Ablation Study:**
   A central claim of the paper is the superiority of the "**layer-aware**" injection strategy (semantics to deep, structure to shallow). However, Table 4 is fundamentally incomplete. To prove that layer separation is superior to unified injection, the authors must compare against a baseline that injects **both** semantic and structural priors into **all** layers (both shallow and deep). Without this "all-layer injection" baseline, Table 4 fails to substantiate the paper’s core architectural claim.

4. **Limited Methodological Innovation for Top-tier ML Conferences (Suitability for ICML):**
   The core academic contribution relies on the heuristic that shallow layers of a U-Net handle local/geometric information while deep layers handle global semantic information. This is a fundamental and widely known property of CNNs/U-Nets in computer vision, exploited extensively in literature ranging from Perceptual Losses to Feature Pyramid Networks. Routing conditional embeddings in a diffusion model based on this established heuristic feels more like **architectural engineering** than a novel machine learning methodology. The theoretical or algorithmic contribution does not quite reach the bar for ICML.

[1] Complexity experts are task-discriminative learners for any image restoration. CVPR 2025.

---

> ### Author Rebuttal · Authors · 2026-03-30
>
> **For Weakness-1 and Question-1:** We thank the reviewer for the valuable suggestion. More precisely, this work belongs to all-in-one multi-task restoration rather than restoration under strictly defined coupled degradations. The single-label cross-entropy in Eq. (14) is only used for degradation-aware representation learning and does not model multiple coupled degradations. Our original intention was to use the complexity of different degradation scenarios to motivate the necessity of unified restoration, but the current wording may overextend the task boundary. We will revise the task definition to avoid confusion with coupled degradation restoration. At the same time, the current degradation classifier based on mutually exclusive cross-entropy cannot handle multiple coupled degradations, which is a limitation of the present work. This paper mainly focuses on unified restoration under multiple single-dominant degradation types and unknown degradation scenarios; therefore, we conduct experiments on nine single-degradation tasks, a 5D all-in-one task, and two unknown-degradation datasets. For real coupled degradations, a more suitable treatment deserves further investigation. We will further clarify this limitation and discuss extending single-label degradation to multi-label modeling as a potential future direction.
>
> **For Weakness-2 and Question-2:** We thank the reviewer for the reminder. Regarding the issue that the improvement from Model C to Ours in Table 3 is only 0.02 dB in PSNR, we would like to clarify that the purpose of this module is not merely to improve the average PSNR but to provide explicit structural constraints for the diffusion process so as to enhance the structural consistency of the restored results. We will supplement a clearer analysis to further explain its role. In addition, we thank the reviewer for mentioning other related methods. MoCE-IR and TPGDiff focus on different aspects. MoCE-IR emphasizes complexity-aware expert routing and efficient inference, whereas the core of TPGDiff lies in hierarchical triplet-prior guidance. In the revised manuscript, we will add a discussion of MoCE-IR and more clearly distinguish the design objectives of the two methods. Meanwhile, the term “parameter-efficient” in Line 257 only refers to the design of the structural token, and we will also rephrase it to avoid ambiguity. Finally, we have supplemented the complexity analysis under a unified experimental setting. As shown in Table 1, all results are evaluated on an NVIDIA Tesla A100 (40G). Although TPGDiff, as a diffusion-based method, incurs a higher inference cost, its computational overhead is still within a reasonable range among diffusion-based methods while achieving favorable restoration performance and generalization ability.
> |Table1|Params|FLOPs|Inference Time|
> |---|---|---|---|
> |Restormer|26M|155G|0.25s|
> |AirNet|9M|301G|0.57s|
> |ProRes|371M|139G|0.21s|
> |PromptIR|36M|172G|0.36s|
> |InstructIR|33M|292G|0.33s|
> |CyclicPrompt|181M|230G|0.47s|
> |DA-RCOT|41M|262G|0.63s|
> |MoCE-IR|25M|142G|0.27s|
> |VLUNet|123M|191G|0.53s|
> |IR-SDE|137M|379G|4.61s|
> |DA-CLIP|295M|138G|4.24s|
> |Ours|355M|189G|4.18s|
>
> **For Weakness-3 and Question-3:** We further supplement two additional ablation settings: (a) injecting the structural prior into both shallow and deep layers and (e) injecting both semantic and structural priors into both shallow and deep layers. All experiments are conducted under the same settings and trained on the Rain200L and Snow100K-L datasets. As shown in Table 2, the performance of scheme (e) instead degrades, indicating that simultaneously injecting semantic and structural priors into both shallow and deep layers not only fails to bring gains but also introduces interference. (Note: If scheme (e) injects semantic information into the shallow layers through an attention-based manner, it leads to OOM under the same experimental setting. Therefore, the actual implementation adopts an adapter-based manner consistent with the structural prior.)
> |Table2|Sem-Deep|Sem-Shallow|Stru-Deep|Stru-Shallow|PSNR↑|SSIM↑|LPIPS↓|
> |---|---|---|---|---|---:|---:|---:|
> |(a)|×|×|√|√|31.73|0.912|0.061|
> |(b)|√|√|×|×|32.38|0.918|0.051|
> |(c)|×|√|√|×|32.18|0.915|0.052|
> |(d)|√|×|×|√|32.63|0.921|0.045|
> |(e)|√|√|√|√|31.51|0.910|0.061|
>
> **For Weakness-4:** Different from existing methods that uniformly inject conditional information, TPGDiff injects different priors into more suitable layers and diffusion stages in a targeted manner rather than applying unified injection, thereby reducing interference among priors and improving restoration performance. The related ablation results also show that the matching relationship between different priors and layers can significantly affect performance. In the revised manuscript, we will express this point more accurately to avoid ambiguity.

---

> > ### Author Rebuttal · Reviewer_5RxG · 2026-04-03
> >
> > The rebuttal is helpful and clarifies several implementation/experimental details. I appreciate the added evidence. However, the main concerns that affected my original overall assessment — especially regarding the motivation of this work — remain only partially resolved. Therefore, I keep my overall score unchanged.

---

> > > ### Author Response · Authors · 2026-04-05
> > >
> > > We sincerely thank the reviewer for spending time and effort reviewing our work and for providing constructive suggestions.
> > > Regarding the lack of clarity in the expression of the motivation, we would like to further clarify that the innovative motivation of this paper mainly targets a long-standing yet insufficiently considered issue in all-in-one image restoration. Existing methods usually regard different priors as conditional information that can be handled in a unified manner, but in fact, heterogeneous priors are not the same in terms of information attributes, granularity of effect, and the network levels at which they are suitable to take effect. What we are truly concerned with is not merely whether priors are introduced but why different priors should not be treated uniformly and which restoration demands they should respectively serve, which is also the core starting point of the method design in this paper.
> > >
> > > Based on this understanding, we believe that this issue is particularly important in image restoration because restoration is different from general generation tasks: it not only requires the results to be visually plausible but also requires preserving the content consistency and structural fidelity of the input image as much as possible while suppressing hallucinations of diffusion models. Therefore, high-level content constraints and low-level structural constraints essentially correspond to restoration demands at different levels; if a unified prior and unified injection manner are still adopted, these constraints are prone to interfere with each other, making it difficult to simultaneously achieve content consistency and local structural fidelity.
> > >
> > > Based on the above motivation, this paper does not treat all priors uniformly but instead coordinates them in a matching manner according to their information attributes and restoration objectives: semantic priors are applied to deep layers to constrain global content consistency, structural priors are applied to shallow layers to preserve local geometric details, and degradation priors are used to perform degradation-aware modulation throughout the entire denoising process. Therefore, the main contribution of this paper is not merely the introduction of multiple priors but rather pointing out that heterogeneous priors should not be treated uniformly and proposing a hierarchical coordination manner that matches different restoration objectives.
> > >
> > > In addition, we provide multiple supplementary analyses (https://anonymous-author2026.github.io/tpgdiff-re2/) to further support the design motivation of this paper.
> > >
> > > The semantic representation analysis (Figure 1) shows that the semantic representations directly extracted from degraded images (T-HQ vs T-LQ) exhibit a clear deviation from the clean semantic space, whereas the distilled student representations (T-HQ vs S-LQ) show higher consistency on the diagonal entries, indicating that they can more stably approach the clean semantic anchor provided by HQ images.
> > >
> > > The time-conditioning analysis of the degradation prior (Figure 2) shows that the influence of degradation information on the denoising process exhibits clear stage-wise variation and is therefore better suited as a dynamic modulation signal throughout the entire diffusion trajectory rather than being used only as a static conditional input, enabling the restoration strategy to adapt to the current noise stage.
> > >
> > > The structural prior analysis (Figure 3) further shows that different structural priors focus on different spatial information and can therefore provide complementary rather than redundant geometric constraints; combined with the pixel-wise error maps, it can be further observed that these structural priors help improve structural errors at different levels, thereby better preserving the geometric consistency and detail fidelity of the restored results.
> > >
> > > We thank the reviewer again for the valuable comments and hope that the above clarifications can more clearly explain the core motivation and method design logic of this paper.

---

### Decision · Program_Chairs · 2026-04-30

**Decision:**

Accept (regular)

**Comment:**

This paper proposes a triple-prior guided diffusion network for unified image restoration. The paper originally received 2xWeakReject and 2xWeakAccept. The main concerns include unclear motivation, missing efficiency evaluation, robustness of semantic priors to image quality, unclear contribution of each prior, etc. The authors have provided rebuttals and three reviewers mention that most of their concerns have been well addressed. One reviewer still questions about the motivation of this work. The authors have responded in detail in the discussion phase. Considering the rebuttal and discussions from all reviewers, ACs recommend accepting this paper. The authors are suggested to carefully revise the paper and incorporate newly conducted experiments according to the comments and discussions.